Coral histology reveals consistent declines in tissue integrity during a marine heatwave despite differences in bleaching severity

Kruse Elisa 1
Brown Kristen T. 1 2
Barott Katie L. kbarott@sas.upenn.edu 1
1 Department of Biology, University of Pennsylvania , Philadelphia , PA , United States of America
2 School of Biological Sciences, University of Queensland , Brisbane , Queensland , Australia
Banaszak Anastazia
Electronic publication date: 2025 Jan 3
Publication date: 2025
Volume: 13
Electronic Location ID: e18654
Received 2024 Jul 18; Accepted 2024 Nov 17
Copyright: ©2025 Kruse et al.
Copyright year: 2025
Copyright holder: Kruse et al.
License: This is an open access article distributed under the terms of the Creative Commons Attribution License, which permits unrestricted use, distribution, reproduction and adaptation in any medium and for any purpose provided that it is properly attributed. For attribution, the original author(s), title, publication source (PeerJ) and either DOI or URL of the article must be cited.
License URL: https://creativecommons.org/licenses/by/4.0/

Keywords: Biological scales, Coral histology, Montipora capitata, Porites compressa, Coral bleaching, Mortality, Climate change

Funding: Kelson Family College Alumni Society Undergraduate Research Grant University of Pennsylvania Center for Undergraduate Research Fellowships and National Science Foundation (NSF) OCE award 1923743 Funding for this research came from the Kelson Family College Alumni Society Undergraduate Research Grant to Elisa Kruse from the University of Pennsylvania Center for Undergraduate Research and Fellowships and the National Science Foundation (NSF) OCE award 1923743 to Katie Barott. The funders had no role in study design, data collection and analysis, decision to publish, or preparation of the manuscript.

==============================
Marine heatwaves are starting to occur several times a decade, yet we do not understand the effect this has on corals across biological scales. This study combines tissue-, organism-, and community-level analyses to investigate the effects of a marine heatwave on reef-building corals. Adjacent conspecific pairs of coral colonies of Montipora capitata and Porites compressa that showed contrasting phenotypic responses (i.e., bleached vs. not bleached) were first identified during a marine heatwave that occurred in 2015 in Kāne’ohe Bay, Hawai‘ i. These conspecific pairs of bleaching-resistant and bleaching-susceptible colonies were sampled for histology and photographed before, during, and after a subsequent marine heatwave that occurred in 2019. Histology samples were quantified for: (i) abundance of mesenterial filaments, (ii) tissue structural integrity, (iii) clarity of epidermis, and (iv) cellular integrity (lack of necrosis/granulation) on a 1–5 scale and averaged for an overall tissue integrity score. Tissue integrity scores revealed a significant decline in overall tissue health during the 2019 heatwave relative to the months prior to the heatwave for individuals of both species, regardless of past bleaching history in 2015 or bleaching severity during the 2019 heatwave. Coral tissue integrity scores were then compared to concurrent colony bleaching severity, which revealed that tissue integrity was significantly correlated with colony bleaching severity and suggests that the stability of the symbiosis is related to host tissue health. Colony partial mortality was also quantified as the cumulative proportion of each colony that appeared dead 2.5 years following the 2019 bleaching event, and tissue integrity during the heatwave was found to be strongly predictive of the extent of partial mortality following the heatwave for M. capitata but not P. compressa, the latter of which suffered little to no mortality. Surprisingly, bleaching severity and partial mortality were not significantly correlated for either species, suggesting that tissue integrity was a better predictor of mortality than bleaching severity in M. capitata. Despite negative effects of heat stress at the tissue- and colony-level, no significant changes in coral cover were detected, indicating resilience at the community level. However, declines in tissue integrity in response to heat stress that are not accompanied by a visible bleaching response may still have long-term consequences for fitness, and this is an important area of future investigation as heat stress is commonly associated with long-term decreases in coral fecundity and growth. Our results suggest that histology is a valuable tool for revealing the harmful effects of marine heatwaves on corals before they are visually evident as bleaching, and may thus improve the predictability of ecosystem changes following climate change-driven heat stress by providing a more comprehensive assessment of coral health.

Introduction

Coral bleaching occurs when abnormally high ocean temperatures cause corals to lose their symbiotic dinoflagellate algae (Symbiodiniaceae). These instances of elevated ocean temperatures, known as marine heatwaves, are increasing in both frequency and intensity due to increased atmospheric carbon dioxide (Oliver et al., 2018). The increasing occurrence of marine heatwaves has led to multiple global coral bleaching events within the last decade (Heron et al., 2016). Coral bleaching events have major impacts on the coral individual as well as the ecosystem, as coral symbionts provide energy to coral colonies, the building blocks of coral reefs. These impacts range from the changes in community composition (Bellwood et al., 2006), reductions in habitat complexity (Hughes et al., 2018b), and alterations to ecological function such as nutrient and energy cycling (Graham et al., 2006). Economic services of coral reefs are also threatened by coral bleaching, and include fisheries production, coastal protection, tourism, pharmaceutical potential, and more (Costanza et al., 2014). To preserve these valuable ecosystems under a rapidly changing climate, more research is needed to better understand the full effects of recurring marine heatwaves on corals and how coral reefs may respond to anthropogenic stressors in the future.

Many questions remain surrounding the long-term health effects of heat stress on coral colonies and reef-wide changes following multiple bleaching events. Coral bleaching does not always translate to coral mortality, as coral colonies can recover and regain their symbionts once heat stress subsides (Hughes et al., 2018a; Matsuda et al., 2020). Kāne’ohe Bay, Hawai‘ i provides a useful location to address these questions, as the reef system has now experienced four marine heatwaves resulting in mass coral bleaching since 1996 (Bahr, Rodgers & Jokiel, 2017; Brown et al., 2023). In the summers of 2014 and 2015, two successive marine heatwaves occurred, resulting in a range of coral bleaching responses within and between species. Specifically, some coral colonies of the dominant reef-building species Montipora capitata and Porites compressa severely bleached while others remained fully pigmented, despite being directly adjacent to one another and seemingly experiencing the same environmental conditions (Cunning, Ritson-Williams & Gates, 2016; Matsuda et al., 2020; Ritson-Williams & Gates, 2020). Impacts of the 2015 marine heatwave were also apparent in M. capitata several years later, as colonies that had remained pigmented (i.e., bleaching-resistant) and those that had bleached (i.e., bleaching-susceptible) exhibited distinct metabolomic signatures from each other (Roach et al., 2021). In 2019, the corals in Kāne’ohe Bay experienced a third marine heatwave in under a decade (Innis et al., 2021; Yadav et al., 2023; Brown et al., 2023), which resulted in 19% of P. compressa and 23% of M. capitata experiencing moderate to severe bleaching across the bay (Yadav et al., 2023). Furthermore, the 2019 heatwave led to metabolic depression in M. capitata and P. compressa regardless of bleaching phenotype or prior bleaching history, although greater declines in metabolism were observed in the bleaching-susceptible individuals (Innis et al., 2021). The absence of visible bleaching in pigmented corals therefore does not indicate the absence of stress, and more needs to be learned about the consequences of marine heatwaves on both bleaching-susceptible and bleaching-resistant individuals.

Histological examination is an informative method for observing the cellular effects of stress on corals that are not readily apparent at the organismal level. Hematoxylin and eosin (H&E) staining is commonly used in observing diseased coral tissue in both M. capitata (Aeby et al., 2016; Burns & Takabayashi, 2011; Work & Meteyer, 2014) and P. compressa (Domart-Coulon et al., 2006; Sudek et al., 2012). Yet, few studies have observed the impacts of heat stress at the tissue-level for these species. Among related genera, tissue analysis during heat stress revealed cellular changes within the host tissue even before a decline in symbiont abundance (Ainsworth et al., 2008). In addition to symbiont loss, a common observation in heat stressed corals is the loss of energetically-costly structures, including mesenterial filaments and epidermal thickness (Brown, Le Tissier & Bythell, 1995; Hayes & Bush, 1990; Szmant & Gassman, 1990; Traylor-Knowles, 2019). In addition, overall poor staining uptake is associated with host cell necrosis and lack of tissue integrity due to heat stress (Traylor-Knowles, Rose & Palumbi, 2017). Further, H&E staining can be used to observe reproductive features and determine potential trade-offs during environmental stress (Henley et al., 2022; Sudek et al., 2012). As such, histological examination of bleaching-susceptible and bleaching-resistant corals throughout a marine heatwave may provide important information about the health and integrity of corals with contrasting bleaching phenotypes at the tissue and cellular levels.

Here, we used histology to determine the impact of the 2019 marine heatwave on coral health for two reef-building species from Kāne’ohe Bay, M. capitata and P. compressa. We examined the responses of individuals from each species with contrasting bleaching phenotypes during the 2015 marine heatwave (i.e., bleached versus pigmented; Matsuda et al., 2020) to the 2019 marine heatwave and the following recovery period. In addition, we compared the tissue-level responses with colony-level bleaching severity and partial mortality as well as reef-wide changes in bleaching prevalence and live coral cover. This study thus compares the effects of the 2019 marine heatwave across multiple biological scales, taking a holistic approach to reef monitoring to better understand the future of coral reefs under global change.

Materials & Methods

Individual colony bleaching and mortality quantification

This study was conducted in the southern portion of Kāne’ohe Bay, Hawai‘ i at Patch Reef 13 (21.4515°N, 157.7966°W). Kāne’ohe Bay is a system of fringing and patch reefs that experiences slightly higher water temperatures compared to the surrounding ocean (Bahr, Jokiel & Rodgers, 2015). Patch Reef 13 has relatively high coral cover, and is dominated by two coral species: Montipora capitata (20–40% cover) and Porites compressa (60–80% cover; Matsuda et al., 2020). Adjacent individuals of bleaching-resistant and bleaching-susceptible corals (conspecific pairs) were first identified during the 2015 bleaching event in which the bleaching-resistant colony remained fully pigmented throughout the heatwave and the neighboring bleaching-susceptible colony showed severe bleaching (Matsuda et al., 2020). The pairs followed for this study included the same 20 colonies of M. capitata and 20 colonies of P. compressa (10 phenotype pairs of each species) assessed by Innis et al. (2021) and Brown et al. (2023), which were photographed every few months from June 2019 through March 2022, incorporating periods of both heat stress during the 2019 marine heatwave and subsequent recovery. The conspecific pairs retained their relative bleaching susceptibility during the 2019 heatwave, such that bleaching-resistant corals in 2015 remained bleaching resistant (i.e., pigmented) in 2019 and vice versa (Innis et al., 2021; Brown et al., 2023). All images were analyzed for bleaching severity by Innis et al. (2021) and Brown et al. (2023), in which individual colonies were categorized as: (1) no signs of paling (0% of the colony); (2) mild paling (<20%); (3) moderate bleaching (20–50%); (4) severe bleaching (50–80%); and (5) fully bleached (80–100%). Bleaching scores used in this study include three timepoints: before the heatwave (July 19, 2019), during the heatwave (October 2, 2019), and after the heatwave (March 9, 2022) to coincide with histology sampling (see below). Partial mortality of each colony from 2019 to 2023 was determined to the nearest 20% by Brown et al. (2023).

Coral histology

Samples of the same coral colony pairs (N = 26–31 colonies per time point) were collected at three time points: (1) before (July 19, 2019), (2) during (October 2, 2019), and (3) after (March 9, 2022) the 2019 marine heatwave. Kāne’ohe Bay is a permanent field site under the Hawai‘ i Institute of Marine Biology which requires permits for coral collection. Coral samples were collected under permits from the State of Hawai‘ i’s Division of Aquatic Resources (permit: SAP 2020-41). Approximately 1 cm3 fragments were cut from sections consistent with the majority of the colony and were immediately fixed in 4% paraformaldehyde at 4 °C for 24 h, then stored in 70% ethanol. Two years following collection, all samples were decalcified in Ca2+-free S22 buffer with 0.5 M EDTA. Samples collected in 2019 were dehydrated, embedded, sectioned, and stained at the University of Pennsylvania. Dehydration was performed with Citrisolv and Safeclear II, xylene substitutes, followed by parafilm embedding using the KD-BMII Tissue Embedding Center. Embedded samples were then sectioned at 10 µm thickness with a microtome (KD-1508A rotary microtome, Kedee), stained with Richard-Allen Scientific Modified Harris Hematoxylin and Eosin-Y (H&E) stains, and mounted using Fisher Chemical Permount mounting medium. A total of 12 samples were also stained with Phloxine B (CAT: 189470250, Thermo Fisher Scientific, Waltham, MA, USA), however, no additional features (e.g., dead/fragmented cells) were revealed using this stain. Samples collected in 2022 were sent to Pacific Pathology for embedding, sectioning, and H&E staining following decalcification at the University of Pennsylvania as described above. Photographs of all H&E stained tissues were taken on an Olympus CX23 microscope with a Retiga R3 CCD camera (Meyer Instruments, Houston, TX, USA) and scaled using the program OCULAR (version 2.0, Advanced Scientific Camera Control) with scale bars calibrated to measurements on the Improved Neubauer cytology slide (Hausser Scientific, Horsham, PA, USA). Histology images displayed in Figs. 1B–1D and 2B–2D were white balanced in ImageJ (version 2) according to the methods in (Sedgewick, 2017) to standardize brightness and reduce glue imperfections. Each slide contained four sections of which the most complete section was selected for representative imaging. These sections were quantitatively scored on a scale of 1–5, in which 5 appeared healthy (e.g., intact tissue structure, presence of mesenterial filaments) and 1 showed stress (e.g., severe necrosis) across each of four characteristics: (i) abundance of mesenterial filaments, (ii) tissue structural integrity, (iii) clarity of epidermis, and (iv) cellular integrity (lack of necrosis/granulation). Sections that scored 1 in all categories showed no mesenterial filaments, very poor or no structural integrity, no clear epidermis, and severe granulation or poor staining uptake. An average was taken across all four tissue parameters to create an overall score of tissue integrity. This scoring method was adapted from Vargas-Ángel et al., 2007 in which each element has been shown to deteriorate as a stress response for corals of the same or closely related species (Brown, Le Tissier & Bythell, 1995; Downs et al., 2009; Traylor-Knowles, Rose & Palumbi, 2017; Work & Meteyer, 2014). Additionally, if reproductive features (e.g., spermatocytes and oocytes) were observed, they were scored as 1 (present) or 0 (absent). Oocytes were counted, their diameter was measured, and their abundance was normalized over tissue section area (calculated in ImageJ version 2). Oocyte development stage was determined by visual features and oocyte size as described in Padilla-Gamiño et al. (2011).

Figure 1 Representative underwater and histology images from a single pair of bleaching-resistant and bleaching-susceptible M. capitata colonies.

(A) Underwater images of a representative pair of adjacent bleaching-susceptible (left) and bleaching-resistant (right) Montipora capitata colonies before, during and after the 2019 marine heatwave. (B–C) Corresponding histological images of H&E stained M. capitata tissues from the bleaching-susceptible colony at the same time point. Scale bars represent 0.25 mm and black arrows indicate the epidermal layer, with surrounding seawater to the top of the image and tissue below. (D) Representative images of mesenterial filaments which appear as darker, more concentrated groups of cells within the tissue (red arrows). Top right image scale bar represents 5 µm; and the bottom right scale bar represents 0.25 mm. Tissue images from the bleaching-susceptible colony during the heatwave did not capture any clear mesenterial filaments. Underwater images and tissue samples were taken before the heatwave (July 2019; top row), during the heatwave (October 2019; middle row), and two and a half years after the heatwave (March 2022; bottom row).

Figure 2 Representative underwater and histology images of a single pair of bleaching-resistant and bleaching-susceptible P. compressa colonies.

(A) Porites compressa colony images of adjacent bleaching-susceptible (left) and bleaching-resistant (right) pairs. The blue ruler seen in “During the heatwave” and “After the heatwave” images represents 1 foot. (B–C) Corresponding histological images of H&E stained P. compressa tissues from the bleaching-susceptible colony at the same time point. Scale bars represent 0.25 mm and black arrows indicate the epidermal layer, with surrounding seawater to the top of the image and tissue below. (d) Representative images of mesenterial filaments which appear as darker, more concentrated groups of cells within the tissue (red arrows). Scale bars also represent 0.25 mm. Images and samples were taken before the heatwave (July 2019; top row), during the heatwave (October 2019; middle row), and two and a half years after the heatwave (March 2022; bottom row).

Reef-wide bleaching prevalence and benthic community composition

Benthic cover was quantified at the same time that photographs and tissue samples of the colony pairs were taken. Quadrats (0.33 m2) were placed every 2 m along a 40 m transect at 1 and 3 m depths (n = 2 transects per depth), totaling ∼80 images per time point, as previously described (Matsuda et al., 2020; Innis et al., 2021; Brown et al., 2023). Benthic community composition was measured through manual identification via CoralNet using 50 randomly allocated points per quadrat (Beijbom et al., 2015). Benthic categories consisted of 23 functional groups or species. The two focal species of this study (M. capitata and P. compressa) were further scored as: (1) pigmented, in which the coral showed no signs of color loss, (2) pale, in which some pigmentation was lost, or (3) bleached, in which the coral fragment was completely white. Additionally, bleaching prevalence was calculated by combining the pale and bleached categories of each species and dividing them by the total benthic cover for each species. Coral cover and coral bleaching prevalence from 2019–2022 were previously published in Brown et al. (2023). Benthic community composition from 2019–2022 is presented here for the first time.

Statistical analysis

All statistical analyses were done using R software (version 4.2.2; R Core Team, 2022). Tissue integrity and bleaching severity data were analyzed using linear mixed effects (lme) models from the nlme package (Pinheiro, Bates & R Core Team, 2022), which were used to determine significance between time (three levels: before, during, after), species (two levels: Montipora capitata, Porites compressa), and phenotype (two levels: bleaching-resistant, bleaching-susceptible), with colony genet as a random effect. Significant interactions were further investigated using pairwise comparisons of estimated marginal means (emmeans package) with Tukey HSD adjusted p-values (Lenth, 2023). Differences in coral colony tissue health were also analyzed using permutational multivariate analysis of variance (PERMANOVA) and principal components analysis (PCA), combining all tissue integrity parameters. For visualization on the PCA, necrosis/granulation scores were inverted such that 5 was high necrosis/granulation and 1 was no necrosis. Resemblance matrices were obtained using Bray–Curtis dissimilarity and 9,999 permutations. A glm of the family “binomial” was used to test differences in the presence (1) or absence (0) of reproductive features within each species across phenotypes along with a glm of the family “Poisson” to test differences in oocyte diameter, both with colony genet as a random effect.

Partial mortality was analyzed using another lme model with species as an explanatory variable, time and phenotype as interacting variables, and colony genet as a random effect. Three Pearson’s product-moment correlation tests were used to compare health data across biological scales by testing for significant correlations between tissue integrity and bleaching scores, partial mortality and bleaching scores, and partial mortality and tissue integrity. The first correlation test compared tissue integrity and bleaching scores from all three time points (before, during and after the heatwave). Partial mortality and bleaching scores were compared using maximum bleach scores from May 2019–March 2022 and a cumulative mortality score for each colony from July 2019–September 2023. The third correlation test compared maximum tissue integrity scores and the same cumulative partial mortality scores for each colony.

Generalized linear models (glm) were built using the family “gamma” for continuous non-normal data (stats package R Core Team, 2022), to determine reef-wide changes in benthic cover across time for each target category (M. capitata, P. compressa, macroalgae, and dead coral). Similarly, a lme was used to determine significant differences before, during, and after the heatwave for reef-wide bleaching prevalence of each species with transect treated as a random effect. All linear models met assumptions of normality and homogeneity of variance, confirmed by graphical analyses of residual plots.

Results

Tissue-level health: thermal stress revealed across all individuals

Analysis of tissue integrity revealed signs of heat stress two months prior to and during the 2019 heatwave in bleaching-susceptible and bleaching-resistant phenotypes of both species (Figs. 1 and 2). Notably, histology revealed poor tissue integrity in both species during the heatwave regardless of previous (2015) or concurrent (2019) bleaching severity. This was evident in histology images during the 2019 heatwave that showed disjunction in tissue structure, indistinguishable epidermal layer, granularity, fewer mesenterial filaments, and poor staining uptake in both bleaching-resistant and bleaching-susceptible colonies (Figs. 1B–1D, 2B–2D). Such features were also evident prior to the start of the heatwave (July 2019), but less severe compared to during the heatwave (October 2019). Coral tissues from two and a half years following the 2019 heatwave showed the most intact tissue, with clear structures and darker stain (Figs. 1 and 2, bottom rows). Species-specific differences included a more pronounced epidermal layer in M. capitata compared to P. compressa, although both species showed similar responses to the heatwave at the tissue-level (Figs. 1 and 2).

Principal component analysis (PCA) of histological parameters revealed significant differences in tissue integrity over time in M. capitata (PERMANOVA; pseudo-F = 92.2; p < 0.0001) and P. compressa (pseudo-F = 120.2; p < 0.0001; Fig. 3). Between bleaching-resistant and bleaching-susceptible phenotypes there were no significant differences for M. capitata (p > 0.57), yet nearly significant differences for P. compressa (p = 0.09; Fig. 3), indicating that tissue integrity was changing over time regardless of bleaching phenotype in M. capitata, but phenotype may have played a role in tissue integrity throughout the heatwave in P. compressa. For M. capitata, the first two principal component (PC) axes explained 86.2% and 7.1% variation, respectively, with clear separation across time and similar variation within each time point (Fig. 3A). For P. compressa, the first two axes explained 93.2% and 3.0% variation, respectively, with some overlap before and after the marine heatwave (Fig. 3B). Overall the PCA revealed less variation in P. compressa traits between time points compared to M. capitata, suggesting that P. compressa colonies showed fewer changes at the tissue-level throughout the 2019 marine heatwave compared to M. capitata.

Figure 3 Principal component analysis of coral tissue health across all tissue metrics.

(A) Montipora capitata and (B) Porites compressa over time, where bleaching-resistant (circle) and bleaching-susceptible (triangle) phenotypes are denoted by shape of the symbol. For both species, the individual effect of time was significant (p < 0.0001). Parameter labels appear in the same vertical order as their respective arrows, where MF: mesenterial filaments. Percent explained variation of PC1 and PC2 is displayed on the x- and y-axes, respectively.

Quantitative analysis that incorporated structural integrity, clarity of epidermis, presence of mesenterial filaments, and level of necrosis/granularity into one metric revealed that mean tissue integrity was greatest two years after the heatwave (4.44 ± 0.13 for M. capitata and 4.59 ± 0.10 for P. compressa), and was lowest during the heatwave (1.52 ± 0.08 for M. capitata and 1.17 ± 0.06 for P. compressa; Fig. 4B). A loss in tissue integrity was also evident before the heatwave, where mean scores for M. capitata were 3.43 (± 0.16) and P. compressa were 3.18 (± 0.90). Notably, the colony images of M. capitata from before the heatwave (July 2019) showed clear bleaching among the susceptible colonies, but relatively intact tissues (Figs. 1A–1C). Statistical analysis of tissue integrity scores revealed significant differences between time (X2 = 166.1, p < 0.0001) and no significance of species, phenotype, or the interaction between time and phenotype (p > 0.30; Fig. 4B). Pairwise analysis showed significance between each pair of dates (p < 0.001), with the tissue integrity most degraded during the 2019 heatwave and most intact two years later (Fig. 4B).

Figure 4 Coral health metrics measured across biological scales before, during and after a marine heatwave.

(A) Conceptual representation of biological scales represented in the study from smallest to largest: tissue-, organism-, and community-level. (B) Overall tissue integrity (mean ± SE) and (C) bleaching severity of colony pairs (mean ± SE) separated by coral species and phenotype from before (July 19, 2019), during (October 2, 2019), and after (March 9, 2022) the 2019 marine heatwave. Large points indicate group means ± SE and smaller points indicate individual genotypes (n = 5 –11), with bleaching-susceptible in gray and bleaching-resistant in black. (D) Bleaching prevalence measured from reef-wide transects (n = 4) as percent pale or bleached coral from total coral cover of each species. Box-and-whisker plots indicate medians and interquartile ranges. Insets indicate statistical significance (∗∗∗p < 0.0001) of individual and interactive effects for time (T), phenotype (P), and species (S) as determined from linear mixed effects models.

Reproductive features in both species were observed in the tissues from before (July 2019) and after the heatwave (March 2022; Fig. S1), although there were no significant differences in the presence of gametes among species or phenotype (p >  0.10). Notably, within hermaphroditic M. capitata there were no significant differences between bleaching-resistant and bleaching-susceptible groups in oocyte abundance per mm2 (p > 0.90) or oocyte size (p > 0.48). Similarly, gonochoric P. compressa did not show significant differences between phenotypes in oocyte abundance (p > 0.98) and oocyte size (p > 0.70), although there were only 13 colonies with oocytes and 17 total oocytes observed (Table 1). Interestingly, there were differences between the two species in reproductive capacity. From the March 2022 time point, M. capitata colonies had a greater number of oocytes present compared to P. compressa colonies, with a mean 1.23 (± 0.27) oocytes per mm2 of tissue compared to 0.28 (± 0.14) oocytes per mm2 of tissue among P. compressa (Table 1). Oocytes from M. capitata were also larger than those from P. compressa, with mean diameters of 65 (± 2.9) µm among M. capitata compared to P. compressa, which had a mean diameter of 31 (± 3.1) µm. Further, M. capitata had oocytes in both Stage I and Stage II of development, whereas P. compressa only contained oocytes at Stage I (Fig. S1). No spermatocysts were identified in either species.

Table 1 Presence, abundance, and size of oocytes in bleaching-susceptible and bleaching-resistant colonies of Montipora capitata and Porites compressa.

Percent of individuals with oocytes determined by the number of individuals with oocytes over the total number of colonies for each group. Oocyte abundance is normalized over each sections’ surface area for mean oocytes per mm 2. SE is the standard error of the preceding metric.

Species	Phenotype	Number of colonies (N)	Number of oocytes present (n)	Percent of individuals with oocytes	Mean oocytes per mm2	SE	Mean oocyte diameter (µm)	SE	
Montipora capitata	Resistant	7	37	71.4	1.26	0.44	71	4.3	
Montipora capitata	Susceptible	10	34	60.0	1.19	0.37	59	3.6	
Porites compressa	Resistant	7	9	42.9	0.28	0.19	35	4.7	
Porites compressa	Susceptible	6	8	33.3	0.29	0.23	36	3.2	

Organism-level health: phenotype-specific responses to the heatwave

There was a significant three-way interaction between species, phenotype, and time on colony-level bleaching severity from the three dates (X2 = 13.61, p = 0.001; Fig. 4C). Bleaching-susceptible colonies of M. capitata and P. compressa had significantly different bleaching severities between species during the heatwave (t.ratio = 7.6, p < 0.0001) as well as just before the heatwave (t.ratio = 6.1, p < 0.0001), with bleaching-susceptible M. capitata exhibiting more severe bleaching than bleaching-susceptible P. compressa. However, there were no significant differences between species for the bleaching-susceptible corals two years after the heatwave when pigmentation had recovered (p = 0.43). Bleaching-resistant corals did not show significant differences between species at any time point (p > 0.16), as these corals remained pigmented throughout the time series for both species. The two phenotypes of M. capitata showed significant differences from each other before and during the heatwave (before: t.ratio = 5.2, p < 0.0001; during: t.ratio = 10.9, p < 0.0001), while P. compressa only showed significant differences in bleaching responses between the two phenotypes during the heatwave (t.ratio = 3.8, p = 0.0004; Fig. 4C). Overall, bleaching-susceptible M. capitata exhibited significantly higher bleaching severity than bleaching-susceptible P. compressa during the heatwave (t.ratio = 7.6, p < 0.0001; Fig. 4C). Partial mortality of individual colonies showed a significant difference between species (X2 = 12.7, p < 0.0004), which was greater in M. capitata (Fig. S2). Across time, mortality was significantly different (X2 = 6.4, p = 0.42), but there were no significant differences between phenotypes (p > 0.99) nor the interaction between time and phenotype (p >  0.81). Pairwise comparisons revealed species-specific differences in partial mortality at all time points (p < 0.0009). Among M. capitata colonies, partial mortality steadily increased throughout the time series, culminating to a mean 37% (± 5%) tissue loss by September 2023 (Fig. S2).

Comparing health metrics across biological scales revealed relationships between tissue-level and organism-level health (Fig. 4A). Specifically, a significant negative correlation was uncovered between tissue integrity and colony-level bleaching severity, where tissue integrity declined with increased bleaching severity in both species (M. capitata: r = −0.53, p = 0.0004; P. compressa: R = −0.45, p = 0.0048; Fig. 5A). P. compressa had less overall bleaching compared to M. capitata and almost no mortality, resulting in no significant correlation between bleaching severity and partial mortality (r =0.04, p = 0.87; Fig. 5B). Despite having more severe bleaching and mortality, bleaching severity and partial mortality in M. capitata colonies were also not significantly correlated (r = −0.026, p = 0.91; Fig. 5B). When comparing maximum tissue integrity with cumulative partial mortality over the entire time period, M. capitata showed a significant correlation (r = −0.52, p = 0.023), whereas P. compressa tissue integrity was not significantly correlated with partial mortality (r =0.25, p = 0.41; Fig. 5C).

Figure 5 Relationships between bleaching severity, tissue integrity, and survival.

Tissue integrity is negatively correlated with colony bleaching severity in both (A) Montipora capitata and (B) Porites compressa. Linear regressions comparing colony-level bleaching severity (as a maximum score from May 2019–September 2022), with individual (C) Montipora capitata and (D) Porites compressa final partial mortality (percent tissue loss from July 2019–September 2023). Final partial mortality is then compared to maximum tissue integrity scores of each individual colony of (E) Montipora capitata and (F) Porites compressa. Lines represent least-squares regressions with gray shading representing 95% confidence intervals. r and p-values calculated from Pearson’s product-moment correlation tests.

Community-level health: minimal reef-wide impact of the marine heatwave

Bleaching prevalence across the patch reef mirrored colony-level bleaching scores, with P. compressa showing less overall bleaching prevalence than M. capitata (Fig. 4D). Comparisons of reef-wide bleaching prevalence before, during, and after the heatwave revealed a significant interaction between time and species (X2 = 7.2, p = 0.03; Fig. 4D). Specifically, M. capitata had significantly higher bleaching prevalence (38%) than P. compressa (7%) in June 2019 before the heatwave (t.ratio = 4.7, p = 0.0004), which then increased to 49% and 38% bleaching prevalence for M. capitata and P. compressa, respectively, during the heatwave (October 2019; Fig. 4D). These patterns indicate earlier bleaching responses from M. capitata, but similar reef-wide bleaching prevalence at the peak of the heatwave for both M. capitata and P. compressa. Two years following the heatwave (March 2022), bleaching prevalence returned to low levels, with M. capitata showing 10% bleaching prevalence and P. compressa showing 2% bleaching prevalence. Despite widespread bleaching during the heatwave, there were no significant changes to benthic composition that were detected up to two years following the 2019 heatwave (Fig. S3). For example, there was no significant change in total coral cover of either species over time (M. capitata: X2 = 0.76, p = 0.38; P. compressa: X2 = 0.72, p = 0.40; Fig. S3). Other benthic categories including dead coral and macroalgae were also tested for changes over time, revealing no significant differences (dead coral: X2 = 2.6, p = 0.99; macroalgae: X2 = 16, p = 0.32; Fig. S3).

Discussion

Coral tissue integrity declined during a marine heatwave regardless of bleaching phenotype

All colonies of M. capitata and P. compressa showed decreases in tissue integrity during the 2019 marine heatwave, despite a lack of visible coral bleaching in bleaching-resistant colonies. These results complement physiological and metabolic measurements on these same individuals during the peak of the 2019 marine heatwave, where metabolic depression and declines in photochemical capacity occurred regardless of visually-observed bleaching (Innis et al., 2021). In our study, both bleaching-resistant and bleaching-susceptible M. capitata and P. compressa showed declines in tissue integrity that manifested as poor stain uptake, declines in mesenterial filaments, and reduced clarity of the epidermis. These results align with several earlier studies that found abnormal tissue architecture and poor stain uptake in coral tissue during heat stress (Brown, Le Tissier & Bythell, 1995; Traylor-Knowles, Rose & Palumbi, 2017; Traylor-Knowles, 2019). Both bleaching-susceptible and bleaching-resistant M. capitata were previously found to lose tissue biomass during heat stress, while P. compressa colonies showed the opposite pattern (Innis et al., 2021), yet we did not find species-specific differences in tissue integrity during the heatwave. Further, tissue integrity was negatively correlated with colony-level bleaching across the time series in both species, suggesting that heat stress not only resulted in severe loss of tissue integrity, but was also related to declines in symbiont density. During the 2019 heatwave, only bleaching-susceptible M. capitata showed significant declines in symbiont density (Innis et al., 2021). Significant symbiont loss accompanied by decreased tissue integrity and cell necrosis may explain a switch in trophic strategies from autotrophically produced energy to the catabolism of tissue biomass, which would be expected to manifest as reduced levels of proteins, lipids, and carbohydrates during marine heatwaves (Grottoli, Rodrigues & Palardy, 2006; Rädecker et al., 2021; Schoepf et al., 2015). Interestingly, the effects of heat stress were evident at the tissue-level in M. capitata before the accumulation of heat stress (i.e., degree heating weeks), similar to a study on Acropora aspera (Ainsworth et al., 2008), suggesting that even short-term incursions of temperature stress above the local maximum monthly mean (MMM) negatively influences coral health. Tissue samples collected approximately two years after the 2019 marine heatwave (March 2022) were the most intact, showing little to no signs of tissue stress and the appearance of distinct reproductive features. Whether the observed patterns were a reflection of seasonality in tissue biomass (e.g., Scheufen, Iglesias-Prieto & Enríquez, 2017) cannot be determined; however, we would not expect seasonal declines in biomass in the summer to be accompanied by the severe declines in tissue integrity that were observed during the heatwave.

Reproductive declines may also result from heat stress observable at the tissue-level (Johnston et al., 2020; Rodrigues & Padilla-Gamiño, 2022), although our data were insufficient to quantify these responses. Specifically, oocyte abundance and size did not significantly differ between bleaching-susceptible and bleaching-resistant individuals within either species; however, this may be due to a limited sample size and calls for higher-frequency sampling across gametogenesis. Species-specific differences in oocyte abundance and size were most likely a result of divergent reproductive strategies, as P. compressa is gonochoric (individuals are different sexes) and M. capitata is hermaphroditic, and both species have different reproductive timelines (Henley et al., 2022; Neves, 2000). Sampling across seasons would help tease apart the effects of heat stress from seasonality, and represents an important avenue of future study.

Decreased tissue integrity suggests a host-specific response to marine heatwaves

Although decreased tissue integrity correlated with bleaching severity, there were no significant differences in tissue health between bleaching-susceptible and bleaching-resistant corals. Interestingly, coral tissue integrity of all individuals showed complete recovery two years following the 2019 marine heatwave, indicating repair of tissue damage and recovery from heat stress for all colonies regardless of historical bleaching and symbiont loss during the heatwave. Differences in bleaching susceptibility during marine heatwaves may be explained by the different species of symbionts hosted by the corals, as M. capitata can host symbionts of two genera: Durisdinium and Cladocopium (Cunning, Ritson-Williams & Gates, 2016). Bleaching-resistant M. capitata have been shown to host a mixed community of Cladocopium and Durisdinium with a higher concentration of the heat tolerant Durisdinium, whereas bleaching-susceptible M. capitata hosted only Cladocopium (Drury et al., 2022a). Additionally, following artificial and natural heat stress, corals with mixed communities of symbionts decreased in Cladocopium abundance and shifted towards Durisdinium-only communities (Dilworth et al., 2021). Such shifts indicate a mode of heat tolerance in relation to the symbiont community in addition to the host genotype (Drury et al., 2022b). However, bleaching-resistant corals still exhibited decreases in tissue integrity during the marine heatwave despite hosting more heat tolerant symbionts and appearing to bleach less. Such results indicate a mechanism of heat stress only present in the host that occurs regardless of symbiont community abundance or composition. A similar pattern was observed for P. compressa, which despite hosting only a single species of Cladocopium (C15 ITS2 classification; Putnam et al., 2012), exhibited significant tissue damage during heat stress for individuals of both bleaching phenotypes. Bleaching-susceptible P. compressa showed less severe bleaching during the 2019 marine heatwave compared to bleaching-susceptible M. capitata in 2019 (Fig. 4C; Innis et al., 2021; Brown et al., 2023), even though both groups had bleached severely during the 2015 marine heatwave (Matsuda et al., 2020). Given that all colonies of P. compressa remained relatively bleaching-resistant in the 2019 heatwave regardless of bleaching history, this may indicate acclimatization to increased temperatures in which individuals have gained increased resistance to bleaching during heat stress (Brown et al., 2023). In 2019, bleaching-susceptible P. compressa showed mild bleaching and significant metabolic depression but was not sustained long enough to result in decreases in tissue biomass, as was observed in M. capitata (Innis et al., 2021). However, both resistant and susceptible colonies of P. compressa, all of which showed mild to no pigmentation loss, displayed significant declines in tissue integrity. These results indicate that this species underwent significant tissue stress during the heatwave despite the absence of visible bleaching. Optimistically, histology revealed full recovery in tissue integrity among P. compressa two years after the 2019 heatwave, similar to M. capitata tissue recovery. Regardless of symbiont presence and historical bleaching, these coral colonies displayed acute heat stress at the tissue-level, and were able to recover two years following a marine heatwave.

Coral mortality and reef-wide changes in benthic community composition

Despite significant differences in visually assessed bleaching severity between resistant and susceptible phenotypes of P. compressa during the 2019 heatwave, bleaching severity was mild in susceptible individuals (Brown et al., 2023; Innis et al., 2021). This corresponded with low levels of partial mortality, which were not significantly different between bleaching-resistant and bleaching-susceptible phenotypes (−2% with coral growth vs. 0%, respectively). However, all corals exhibited signs of heat stress at the tissue-level, which may explain the partial mortality that occurred during the years following the heatwave even in the absence of visual signs of bleaching. This response is in contrast with the response of these same individuals to the previous 2015 heatwave, where the susceptible corals bleached severely and had higher partial mortality in the two years following the 2015 heatwave (22%) than they did after the 2019 heatwave (0%) or the resistant corals following the 2015 heatwave (4%; Matsuda et al., 2020). The lower bleaching severity and rates of mortality in bleaching-susceptible P. compressa following the 2019 heatwave suggest these corals may have undergone beneficial acclimatization or experienced less heat stress in the subsequent 2019 event relative to the 2015 event. In contrast, the response to repeated heatwaves differed for M. capitata. In the two years following the 2015 heatwave, these same individuals of M. capitata exhibited 14% and 10% partial mortality among bleaching-susceptible and bleaching-resistant individuals, respectively (Matsuda et al., 2020). Following the 2019 heatwave, M. capitata exhibited greater partial mortality among both phenotypes after the 2019 event (37% for bleaching-susceptible and 36% for bleaching-resistant). This increase in mortality following the second event for M. capitata indicates that these individuals may be accumulating the negative effects of heat stress (i.e., incomplete recovery), and thus performing worse upon repeat exposure to heat stress. Across both species, decreased tissue integrity was correlated with increasing bleaching severity, supporting the hypothesis that heat stress can be seen at the tissue-level in addition to visual signs of bleaching. Interestingly, partial mortality was not significantly correlated with increasing colony bleaching severity for either species. Instead, partial mortality correlated with decreased tissue integrity in M. capitata, although they did not correlate in P. compressa. This may indicate that for some species, such as M. capitata, tissue integrity may be a better predictor of partial mortality than colony bleaching severity. These results support the importance of combining multiple physiological metrics across biological scales to better predict coral mortality rates following marine heatwaves.

The 2019 marine heatwave peaked in October 2019 with 5.1 °C-weeks−1 at Patch Reef 13 (Brown et al., 2023), leading to significant loss in tissue integrity, moderate bleaching responses, and partial mortality among all individuals of bleaching-susceptible M. capitata. In comparison to other marine heatwaves observed in Kāne’ohe Bay, such as the bleaching event in 2015 in which some areas of the bay recorded 14 °C-weeks−1 (Brown et al., 2023) and more severe bleaching responses (>40% of all colonies; Bahr, Rodgers & Jokiel, 2017), the 2019 marine heatwave was relatively moderate. This is consistent with our findings, where no measurable changes in coral cover were observed, and only significant bleaching in sensitive individuals of the less thermally-tolerant species, M. capitata. However, partial mortality among the same colonies were observed after the 2015 bleaching event (Matsuda et al., 2020), indicating that the cumulative effects of multiple bleaching events may result in significant mortality over longer periods of time. In its most extreme case, one bleaching-susceptible and one bleaching-resistant M. capitata colony showed complete mortality (100%) from November 2015–September 2023 (Brown et al., 2023). These patterns were also qualitatively observed, where entire sections of reef consisting of M. capitata appeared dead, indicating that significant species-specific mortality may be occurring following bleaching events, even though declines in live coral cover were not captured in our surveys. However, another study using more sensitive sampling methods detected declines in live coral cover of 19% for P. compressa and 23% for M. capitata in Kāne’ohe Bay after the 2019 heatwave (Yadav et al., 2023), corroborating our observations that M. capitata mortality following heatwaves can lead to significant loss of coral cover. These observations call for further analysis of potential ecosystem-wide changes due to a loss in thermally-sensitive coral species such as M. capitata, and accompanying changes in benthic community composition.

Conclusions and future directions

Histological analysis of coral tissues revealed signs of stress in the absence of visual symptoms of bleaching and onset of tissue stress prior to the accumulation of heat stress on the reef (as measured by degree heating weeks) in two distantly related reef-building coral species. These results indicate that histology is a valuable method for detecting coral stress before or in the absence of a visible stress response, and could be a useful tool for predicting coral health and mortality following heat stress. Marine heatwaves pose a major threat to the conservation of coral reefs, yet there is hope that the acclimatization of corals, in conjunction with proper management strategies, continued research across biological scales, and global policy to limit greenhouse gas emissions, can ensure a future for these unique ecosystems.

Supplemental Information

Supplemental Information 1 Representative images of Montipora capitata and Porites compressa oocytes from March 2022

Left image shows a Stage II oocyte with a clear nucleus and darker nucleolus on the left edge of the nucleus. Right image shows a Stage I oocyte from a Porites compressa colony with a faint nucleus and darker nucleolus in the center. Scale bars represent 50 um.

Supplemental Information 2 Colony-level partial mortality over time among bleaching-resistant and bleaching-susceptible groups of both species

(A–B) Tissue loss (mean ± SE) as a percentage of the entire colony compared to its original size in July 2019. Percentages are cumulative and decreased partial mortality represents tissue regrowth. Bleaching-susceptible means are in gray and bleaching-resistant means are in black. Inset indicates statistical significance (∗∗∗p < 0.0001) of species (S) and time (T). Underwater images were taken before (July 2019), during (October 2019), and two years after (March 2022) the 2019 marine heatwave.

Supplemental Information 3 Reef-wide changes in benthic composition from before, during, and after the 2019 heatwave

Each line represents a different species or functional group from transect analysis with means and 95% confidence intervals (transect N = 2–4) expressed as percent cover. Benthic images were taken before (June 2019), during (October 2019), and two years after (March 2022) the 2019 marine heatwave. No significant changes were observed within each species functional group.

We thank Ford Drury of the Coral Resilience Lab for hosting members of the Barott Lab at the Hawai‘ i Institute of Marine Biology during data collection, and the entire Barott Lab for sample processing and coding support. Ben Glass assisted with identification of reproductive features.

Additional Information and Declarations

Competing Interests

Author Contributions

Field Study Permissions

Data Availability

The authors declare there are no competing interests.

Elisa Kruse conceived and designed the experiments, performed the experiments, analyzed the data, prepared figures and/or tables, authored or reviewed drafts of the article, and approved the final draft.

Kristen T. Brown analyzed the data, prepared figures and/or tables, authored or reviewed drafts of the article, and approved the final draft.

Katie L. Barott conceived and designed the experiments, authored or reviewed drafts of the article, and approved the final draft.

The following information was supplied relating to field study approvals (i.e., approving body and any reference numbers):

Coral samples were collected under permits from the State of Hawai’i’s Division of Aquatic Resources (permit: SAP 2020-41). Kane’ohe Bay is a permanent field site under the Hawaii Institute of Marine Biology which only requires permits for coral collection.

The following information was supplied regarding data availability:

The data and code is available at Zenodo: Kruse, E. (2024). Coral histology reveals consistent declines in tissue integrity during a marine heatwave despite differences in bleaching severity [Data set]. Zenodo. https://doi.org/10.5281/zenodo.12746083.

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
