# Peer review of "Coral histology reveals consistent declines in tissue integrity during a marine heatwave despite differences in bleaching severity"

_PeerJ, doi:10.7717/peerj.18654_

## Round 0.1 · original submission · Major Revisions

Two expert reviewers have evaulated your manuscript and their comments can be seen below As you will see both reviewers have made very valuable suggestions for improving the manuscript that should be taken into account in a revised version. Please ensure that you take each comment into account in a rebuttal that clearly shows what changes were made and where they are located.

·

Basic reporting

This paper is very clear, uses proper sources and background, as well as mostly professional figures. It uses relevant results to the hypotheses provided and data are logically used.

Experimental design

Experimental design is well thought out, relevant, and very meaningful to ongoing research. This paper fills a much needed gap!

Validity of the findings

Conclusions are well stated, and experiment and results could be replicated based on the information provided.

Additional comments

This is really well done! I think just a few things to tighten it up would be helpful such as a map to understand the study site and improving your histology image quality. But overall, I'm very excited to see this paper published!

Reviewer 2 ·

Basic reporting

The manuscript is difficult to read in several areas. In particular the methods and results could use more clarity. I have provided more line by line feedback in additional comments below.

Experimental design

More details on the design of the study are needed. In particular the source of the data is not clearly stated. I suspect all but the histology data have been published elsewhere and the methods and discussion do not clearly reflect that. I suspect this is to do with lack of clarity in writing rather than any problems with design, but the authors should outline things better. See additional comments below.

Validity of the findings

It was difficult for me to assess the histological images that they show. This is not an area of specialty to me and the authors didn't provide any arrows or highlight specific portions of the images to help explain their characterizations and classifications. What should we be looking at in the histological sections? While I do not suspect any issues, this is not reader-friendly and could easily be improved in a a revision. See additional comments below.

Additional comments

Kruse et al. Coral histology reveals consistent declines in tissue integrity during a marine heatwave despite differences in bleaching severity
The authors have conducted histological analyses of a set of historical samples (I think, see notes below; it is unclear what is new and what is previously published) of coral colonies that experience multiple bleaching events and differentially bleached. The authors find that the histology is similar regardless of bleaching status. This is an interesting finding that is buried in a lot of information that (I suspect) is published elsewhere and I think the authors could write a more streamlined paper that highlights this finding more clearly. There is valuable data here, I think the paper needs major revisions to be suitable for publication.
In general, I think the paper suffers from a lack of organization and too much other content from other related publications that make the essence of the paper difficult to discern. I suggest:
• Reorganizing the data to make tissue integrity the main focus of all sections (since I believe that is the aspect of the study that is new). For example, the last figure (Fig. 5) presents data that are described in the first section of the methods.
• Limit the focus on the parameters that have been previously published and be more clear about when this occurs. Some of the methods are written as if the authors reanalyzed previously published data, some indicate the data were reused.
• The aspect of organizational scale is lost in the weeds in the paper. Initially, reading the abstract, I thought this approach was interesting and unique. But I wasn’t able to discern patterns across scale or glean a take home message related to scale. The image in Fig. 4a is great but is never discussed or referenced in the paper.

An important note before reporting on the details of the content of the manuscript is the author order is different on the Peer J cover page (pg. 4 of PDF) compared to the manuscript document (pg. 5 of PDF). The order of authors Barott and Brown are different. The authors should clarify author order.

Major comments (in text order)
1. Abstract – details are vague and should be clarified to provide more information for the reader. While I recognize that some of these questions that I have posed pertaining specifically to the abstract are better clarified in the main text, the abstract is not easily understandable without more details and the overall purpose and main point of the study is lost.
a. The description of the study design is not clear (L21-24). I think it would be better to remove mention of results from this sentence and focus on the sequence of events. When were samples taken? How was bleaching severity (L27) and partial mortality (L28) quantified?
b. Similarly, the language of the results sentences need more clarity: for example, what the decline compared to (L24), how much was the decline?, what does ‘tissue integrity’ mean? I think this portion of the abstract would benefit from some numbers to support the statements.
c. What data support the statement about consequences for fitness? (L32) Can some details be included in the abstract? Without them, the statement is speculative at best and doesn’t seem connected or relevant to the other information presented at worst.
d. The last sentence is very broad and does add much interesting content to the manuscript. I suggest they replace it with a broader finding of their study or use the extra word space to incorporate more details of the study as suggested in the points above.

2. Source of data – it is not clear to me whether the values for bleaching severity and colony partial mortality/growth that were used for this study were previously published or were the values reassessed from existing photographs by the authors of the current study. Both interpretations could be assumed from the text in L114-121 and in other areas of the text as outlined below.
a. The description of data source pertaining to benthic cover and reef bleaching severity is more clear and similar statements (L134-137) should be applied to tissue loss and individual colony bleaching status, as appropriate.
b. If previously published, how does the use of the data here differ from the original use of the data?
c. L397-399 – this statement in the discussion sounds like the current study has different data set than the others, so I’m not sure if the data is new or reused. Perhaps they should delete “this study” from L399.

3. Authors should define what they mean by bleaching susceptible/resilient in the methods (around L111). Some description occurs in the abstract and introduction, but I think it is more appropriate to provide a detailed explanation in the methods since this forms the basis for an important factor in their study design.
a. Did the resistant colonies never bleach in 2015 or was there a gradient of pigmentation?
b. Did the bleaching status of colonies change between 2015 and 2019?
c. More details are needed in the methods to justify the category distinctions.
d. If these descriptions are provided in previously published work, the authors should more clearly state they are following prior methods, but should also outline the pertinent details here.

4. Were the LME models in R conducted on the individual data and not the reef-wide data? It would be helpful to state this in the text more clearly. Were any statistical analyses conducted across scales? I am having trouble understanding the sentence in L201-205. It is quite long and it is not clear what is being compared/analyzed/assessed.
a. Overall, for the statistics I think it would be helpful to group the analyses by organizational scale, since that seems to be one of the main contributions of the paper. I think this may be done already, but the authors could be more explicit in how these are described. With a lot of data presented it is difficult to keep track and the statistics section could help remind readers how the multiple pieces fit together.
b. This could then be better mirrored in the results. Again, I think this is happening, but it is difficult to follow. Simpler headings would be helpful. I know the authors are going for nature-style headings that explain significance, but there is so much going on here, that would be easier to focus on tissue scale vs organism scale vs community scale.

5. Histological features described in L219-222 should be pointed out more clearly on Figs. 1 and 2. It is not obvious what the differences are from the images shown (to a non-specialist in histology at least).
a. What is the timing of the images shown in these figures? The text to more specific dates (months and years in some cases) that are not possible to follow from the images shown. For example, “two years” after the heat wave is discussed in the text. Is that shown in the “After” images in the figures?

6. The authors never refer to Kolinski 2004, a UH PhD thesis that compares reproductive biology of these species from Kaneohe Bay and elsewhere around Oahu. I think it would be useful to at least compare oocyte size to that former work as they can provide some insight on whether bleaching is impacting oocyte size in these species and whether there have been any long-term changes in size. I believe the details of oocyte size are in the thesis.

7. Data for SST that the colonies were exposed to during the first bleaching and the subsequent should be included somewhere in the manuscript or the supplemental files. In the discussion, the authors mention that in 2019, P. compressa colonies didn’t bleach as much regardless of the prior status – is this just because the temps in 2019 were not as severe as those in 2015? Having the temperature data incorporated may help explain some of the patterns observed.
a. Some mention of degree heat weeks is included in the next paragraph (L443). But it is not appropriate to compare DHW in different parts of the world to each other to assess severity. Reefs in Hawaii operate very differently from those in Australia. Reefs in Kaneohe Bay operate very differently from those in other parts of Hawaii.

8. The discussion section is the strongest part of the paper, but could use additional organization to make the point of the paper more clear. The discussion also heavily reports on prior work that used these samples, without necessarily expanding on how the histological portion of the study contributes. I think the overall text in this section could be reduced and reorganizedfor easier understanding. The conclusion and future directions section are not needed and can be deleted unless required as a section for PeerJ. I don’t believe the current text there adds to the manuscript.

Minor comments
1. Figures seem out of order with Fig. 4 described before Fig. 3

2. Statement in L233 – the image in Fig 1a, left side looks like a dead colony to me, and not like “clear bleaching” as described by the authors. The colony in the image looks to have lost rugosity which happens when portions are dead. This may be an issue with the image itself or the quality in the PDF.

3. L246-246 – it might be helpful to point out the P. compressa is gonochoric, to help explain why a limited number of colonies had oocytes. I’m not sure if the sexual systems for these species have been described in the text anywhere to this point.

4. Wording of L260-264 is very confusing.

5. L226, 268: should the figure references direct the reader to Figure 3, not Figure 5?

6. Supplemental figures are missing captions.

---

## Round 0.2 · accepted · Accept

Thank you for addressing all of the comments of the reviewers in such a detailed and organized fashion. I am satisfied with the changes that have been made ot the manuscript and find it acceptable for publcation in PeerJ.